

**Industrial and Residential Ground Water Physico-Chemical Properties Assessment in**
**Lagos Metropolis**
**Lekan Taofeek Popoola[1*], Adeyinka Sikiru Yusuff[1] and Tajudeen Adejare Aderibigbe[2]**
*[1*] Unit Operation and Material Science Laboratory, Department of Chemical and Petroleum*
*Engineering, Afe Babalola University, Ado-Ekiti, Ekiti State, Nigeria*
*[2]Chemical Science Department, Yaba College of Technology, Yaba, Lagos State, Nigeria*
*Corresponding author: popoolalekantaofeek@yahoo.com, +2348062397669*
**Abstract**
This study investigated ground water quality collected from two industrial and residential
locations each of Lagos metropolis. Prescribed standard procedures of American Public
Health Association (APHA) were used to measure physico-chemical parameters of each of
the ground water samples which include pH, EC, DO, TDS, BOD, COD, anions ($Cl^-$, $NO_3^-$,
$SO_4^-$, $PO_4^-$) and heavy metals (Cu, Zn, Pb, Mn, Fe, Co, Cd and Cr). From laboratory analysis,
measured physico-chemical parameters were within the permissible ranges specified by
WHO and NSDWQ except pH, TDS, EC, Pb, Mn and Fe for ground water samples from
industrial locations and pH, Pb, Mn and Fe for residential locations. Higher concentrations of
TDS and EC reported for ground water samples from industrial locations were attributed to
heavy discharge of effluents from industrial treatment plants as well as dissolution of ionic
heavy metals from industrial activities of heavy machines. Statistical Pearson's correlation
revealed physico-chemical parameters to be moderately and strongly correlated with one
another at either $p < 0.05$ or $< 0.01$. In conclusion, ground water samples from residential
locations are more suitable for drinking than those from industrial locations.
**Keywords:** Industrial, Ground Water, Residential, Lagos Metropolis, Physico-Chemical

**Introduction**

Lagos has been identified as the most populous mega-city in Nigeria controlling 40% of the
country's industrial and manufacturing activities contributing 8000 tons of hazardous waste
per year into the environment (Adewolu et al. 2009). Due to these attributes, enormous waste
effluents are being generated on an hourly basis through industrial and residential activities
with higher demand for domestic water consumption linked to her densely populated instinct.
However, these effluents are characterized with toxic and hazardous materials containing



dangerous heavy metals which become sediments in ground water by leaching after their
disposal constituting health hazards to Lagos habitants whose major source of water supply
for domestic purposes comes from this origin. Thus, assessment of ground water quality
based on health and safety regulations specification before domestic use is highly imperative.
Many laboratory procedures and tools involving parameters evaluation of ground water
assessment such as pH, acidity, temperature, salinity, turbidity, alkalinity, electrical
conductivity, total soluble solids (TSS), total dissolved solids (TDS), biological oxygen
demand (BOD), chemical oxygen demand (COD), dissolved oxygen (DO), heavy metal
concentration and so on have been applied (Edwin et al. 2015; Rahmanian et. al. 2015;
Dissmeyer 2000). It is believed that estimated parameters with concentrations higher than
those specify by the World Health Organization (WHO) and other health regulatory bodies
suggest poor drinking water quality (WHO 2011). This great challenge has motivated
researchers and governmental agencies around the globe to engage in series of investigations
(Tuzen et al. 2006; Heydari et al. 2012).
Various applicable and efficient techniques of heavy metals removal from industrial effluents
had been published (Gunatilake 2015; Aryal et al. 2015) while factors influencing their
removal had also been presented elsewhere (Chipasa 2013; Piccirillo et al. 2013). Many
laboratory analytical techniques such as inductively coupled plasma and mass spectrometry
(ICP-MS) (Faisal et al. 2014), flame atomic absorption spectrometry (FAAS) (Behailu et al.
2017), direct extraction/air acetylene flame method (Rahmanian et. al. 2015) and graphite
furnace atomic absorption spectrophotometer (AAS-GF) (Mkadmi et al. 2018) had been
applied to evaluate concentrations of heavy metals in ground water samples with different
statistical analytical tools such as principal component analysis (PCA) (Faisal et al. 2014;
Duan et al. 2015), statistical package for social scientists (SPSS) (Lovelyn et al. 2014),
analysis of variance (ANOVA) (Edwin et al. 2015), least significance difference (LSD)
(Sabhapandit et. al. 2010) and single factor analysis of variance (t test) (Shigut et al. 2017) to
analyse the results.
Ground water is the major source of drinking water in Lagos metropolis due to high disposal
of wastes in different forms into water bodies enhanced by her densely populated feature
thereby contaminating other water sources. However, waste effluents from industrial
treatment plants and solid wastes from residential areas find their ways into ground water via
leaching. Thus, ground water quality must be regularly monitored in these locations. In this
study, ground water samples were obtained from prominent industrial and residential

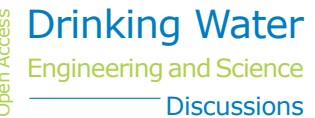

locations of Lagos State, Nigeria and were evaluated to know heavy metals (copper, zinc,
lead, manganese, iron, colbalt, cadmium and chromium) that are present and other physico-
chemical properties such as total dissolved solid, pH, electrical conductivity, chemical
oxygen demand, biological oxygen demand, chloride, nitrate, sulphate and phosphate. The
obtained values were compared with standard values set by Nigerian Standard for Drinking
Water Quality (NSDWQ) and World Health Organization (WHO) guidelines to ensure high
water quality before drinking and other domestic purposes. Statistical Pearson's correlation
was used to check the level of correlation of physico-chemical parameters at $p < 0.05$ or $<$
0.01 in ground water samples collected from examined locations.

**Materials and Methods**
**Study Area**
Lagos has been known as the largest and most populous city in Nigeria with a population of
17.5 million (Adewolu et al. 2009). It is approximately lying on longitude 20 42′E and 32 2′E
respectively and falls between latitude 60 22′N and 60 2′N with 22% of its 3,577 $km^2$ to be
lagoons and creeks. Lagos has 29 industrial estates and 4 central business districts attributed
with 26.7% Gross Domestic Product (GDP) out of Nigeria's total GDP. It has highest
emission level of 8000 tons of hazardous waste yearly, most of which is directly discharged
into the Lagos lagoon.
The case study areas are Deli Foods Nigeria limited, located along Apapa-Oshodi (an
industrial center known with many manufacturing activities) and OK Foods, located at
Ladipo in Mushin area of Lagos whose major productions are biscuits and confectioneries.
Oshodi and Agege community boreholes were chosen as sites for residential ground water
collection. The high population with enormous commercial activities attributes geared
choosing these industrial and residential areas as case studies for this research work.
**Samples Collection**
Two litres each of ground water samples were collected from boreholes of Deli foods (IW1)
and OK foods (IW2).  Also, ground water samples were collected from Oshodi and Agege
community boreholes, each located at 40 km away from Deli foods (RW1) and OK foods
(RW2) respectively. This sample collection exercise was done during May 2018 and samples



were kept in 5 L-capacity plastic kegs rinsed with hexane and distilled water in the laboratory
to remove impurities that may be present before collection. The kegs were instantly covered
with aluminum foil and lids sealed to avoid interference with atmospheric contaminants.
Sample bottles were adequately labeled after which samples were analyzed for different
physico-chemical properties present in the waste water treatment laboratory of Afe Babalola
University, Ado-Ekiti, Ekiti state.
Samples pH were measured using OAKION pH meter (S/N 2202625, Eutech Instruments,
Singapore). Electrical conductivity (EC), dissolved oxygen (DO) and total dissolved solid
(TDS) were calculated by electrometric method. Salinity was determined using ion exchange
electrode method. Calcium present was determined using EDTA method. Iron was
determined using Hach method 8008 (Ferro Ver). Nitrate, sulphate, phosphate and chemical
oxygen demand (COD) were determined using colorimetric method with HACH
standards/methods 8039 high range, 8051, 8190 and 8155 low range respectively.
Colorimetric salicylate method (HACH method 8155 Low Range) was used to calculate
ammonia present. Biological oxygen demand (BOD) was determined using azide
modification method (5210A) prescribed by American Public Health Association (APHA,
2012). Chlorine content was determined using argentometric method while heavy metals
concentrations were calculated using flame absorption spectrophotometer (Buck Scientific
AAS VGP 210 model). All parameters were measured in mg/L with the exception of EC
measured in μS/cm while pH and DO were unitless. Analysis of variance (ANOVA) was the
statistical tool used together with computer SPSS 16.0 windows application.

**Results and Discussion**

Each of the samples collected was analyzed for 23 physico-chemical properties namely pH,
EC, DO, TDS, BOD, COD, nitrate, phosphate, Cl, sulphate, solids salinity, ammonia and
heavy metals which include chromium, nickel, cadmium, lead, cobalt, mercury, copper, zinc,
vanadium, manganese and iron.

**pH**

pH measures the degree of alkalinity or acidity of a solution and calculated by taking
negative logarithm of the hydrogen ion activity. The pH values for water samples obtained
for IW1, IW2, RW1 and RW2 were $7.58 \pm 0.06$, $8.31 \pm 0.02$, $6.35 \pm 0.15$ and $6.46 \pm 0.05$

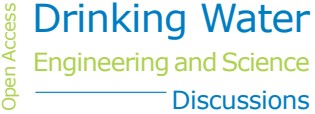



respectively as shown in Figure 1. This revealed pH range of 6.35 to 7.58 with minimum and
maximum exhibited by RW1 and IW1 respectively. The industrial water sample (IW2) was
alkaline while residential borehole water samples (RW1 and RW2) were slightly acidic and
values were not within the permissible range (6.5-8.0) specified by Nigerian Standard for
Drinking Water Quality (NSDWQ) (SON, 2007) and World Health Organization (WHO,
2009). The alkalinity of water sample (IW2) may be attributed to the presence of
bicarbonates, part of essential raw materials for production, lost into the soil and percolates
into the underground soil via rain water. Slightly acidic nature of RW1 and RW2 may result
from the formation of carbonic acid due to the presence of more atmospheric carbon dioxide
dissolution arising from larger population in residential areas than industrial areas (Tiwari et
al, 2015). This may be transported from soil surface level to form deposits in the ground
water via some chemical processes over period of time. Water with high alkalinity has proven
to cause swelling of hair fibres and gastrointestinal irritation (Rose 1986). Acidic water has
been identified to cause damage to cells of mucous membrane, eyes and skin irritation (WHO
1986; Meinhardt 2006). Also, acidic water contributes majorly to corrosion of metals coupled
with disinfection efficiency causing indirect effect on human health.

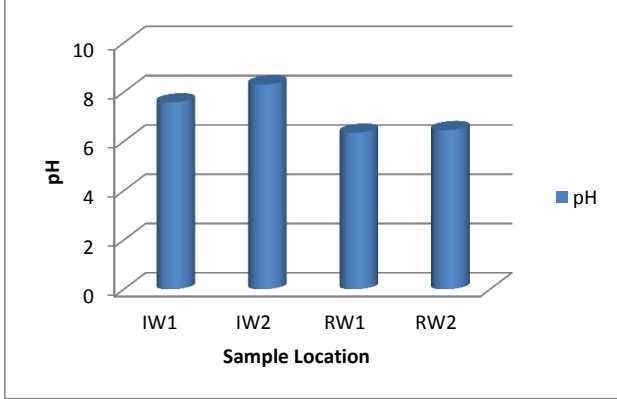

**Figure 1:** pH Spatial Variation of Water Samples





**Total Dissolved Solids (TDS)**

This is a measure of total solids (both organic and inorganic substances) present in water sample either in ionized or molecular suspended form. Filtration of water sample through a medium is usually done before being subjected to high temperature to determine its salinity. The respective TDS obtained for samples IW1, IW2, RW1 and RW2 were 559.2, 589.7, 319.5 and 247.5 mg/L as shown in Figure 2. Only water samples located within industries (IW1 and IW2) revealed maximum TDS values higher than the permissible value (500 mg/L) of NSDWQ and WHO. Minimum variation below permissible value was exhibited by RW1 and RW2. Maximum TDS exhibited by IW1 and IW2 is an indication of saline water which may be attributed to (1) presence of natural solute via dissolution of soils and weathering; and (2) discharge from industrial treatment plants causing soil contamination leaching and point source ground water pollution (Boyd, 1999). Implications of high TDS are (1) organoleptism in human and; (2) reduction in performance of pipes, filters and valves due to scale accumulation (Atekwanaa et al, 2004).

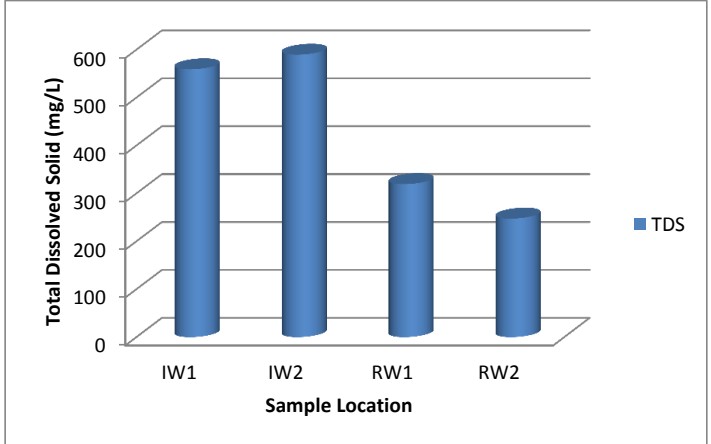

**Figure 2:** TDS Spatial Variation of Water Samples

**Electrical Conductivity (EC)**

EC is directly related to TDS as it measures ionic content of water sample which determines its ability to conduct an electric current. As TDS concentration of water sample increases, the





ionic strength also increases. The values of EC obtained for IW1, IW2, RW1 and RW2 were
1190, 550, 890 and 450 µS/cm respectively as presented in Figure 3. The values range
between 450-1190 µS/cm with IW1 exhibiting maximum EC while RW2 exhibited minimum
EC. All values obtained were below the permissible value of 1000 µS/cm specified by
NSDWQ and 900 µS/cm specified by WHO for drinking water except IW1. The intolerable
EC value exhibited by IW1 could be attributed to (1) dissolution of ionic heavy metals from
industrial activities of heavy machines which later found their ways into ground water via
leaching of sub-soil layers (Eruola et al. 2012); and (2) higher temperature of the location
enhancing movement of ions under electrostatic potential (Oguntona et al. 2012). The side
effects are mainly water corrosiveness of water and heavy metals presence make the water
unsuitable for drinking.

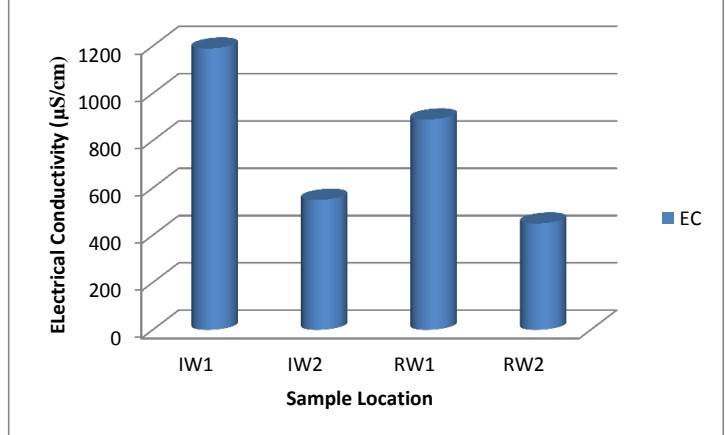

**Figure 3:** EC Spatial Variation of Water Samples

**Biological and Chemical Oxygen Demands (BOD and COD)**
COD measures oxygen requirement for organic matter chemical oxidation to take place via
assistance of strong chemical oxidant while BOD gives a measure of oxygen requirement for
biodegradation of carbonaceous matter in a sample. The values revealed by IW1, IW2, RW1
and RW2 for COD and BOD were 2, 4, 11 and 7 mg/L; and 1.2, 1.8, 4.7 and 2.9 mg/L as
presented in Figures 4(a) and (b) respectively. All values were below the maximum

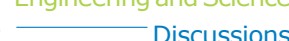

permissible values of 40 and 10 mg/L specified for COD and BOD respectively by WHO and
make them suitable for domestic usage. IW1 revealed minimum COD and BOD while RW1
revealed maximum COD and BOD levels. This is due to sufficiently large volume of
municipal and solid wastes generated within the densely populated region, transported into
the ground via leaching, constituting to water pollution by increasing the organic content
amount (Sumant et al. 2015). Thus, more oxygen is required by the microbes for their
degradation.

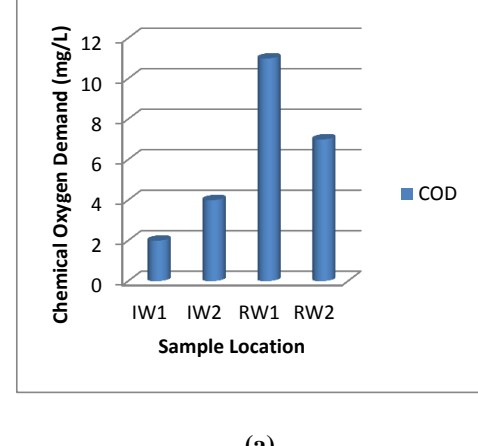
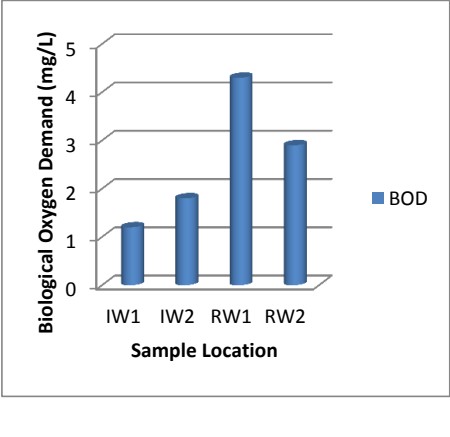

**(a)**                                                    **(b)**
**Figure 4:** Water Samples Spatial Variation for (a) COD   (b) BOD

**Chloride (Cl⁻)**
Concentration of chloride varies from types of water and has been found to exist naturally in
form of sodium and potassium salts. It's a stable water component whose concentration is
uninterrupted by both bio- and physio-chemical processes. As shown in Figure 5, the
concentration of chloride ranges from 52.2 to 88.6 mg/L with RW1 and IW1 having lowest
and highest concentration. All measured concentrations were below the maximum
permissible values of 250 and 600 mg/L specified by NSDWQ and WHO respectively.
Presence of chlorides could be due to (1) chloride-containing soils and rocks undergoing
leaching which later got in contact with underground water for all examined locations
(Aremu et al. 2011) (2) high chloride-rich sewage and municipal effluents discharged by


residents in examined locations for RW1 and RW2 which later found its way into
underground water (Gorde et al. 2013) (3) chloride salts used as essential ingredients for
confectionaries production discharged as industrial effluents in investigated locations for IW1
and IW2. Chlorides have been investigated as essential ingredient for activities involving
human body metabolism (Mohsin et al. 2013). However, excessive chlorides concentration in
water could lead to (1) laxative effect ((2) metallic pipes damage and (3) unsuitability of
water for agricultural irrigation (Raviprakash et al. 1989).

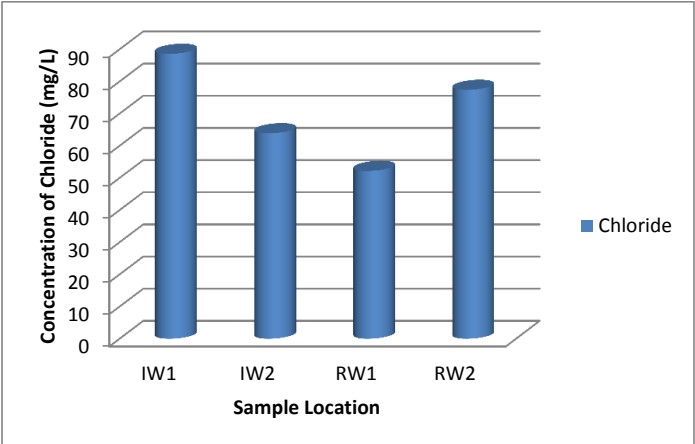


**Figure 5:** Spatial Variation of Chlorides in Water Samples

**Nitrate (NO$_3^-$)**
Nitrate is oxidizing form of N$_2$ compound which can be produced from decaying vegetables,
organic matter of animals, fertilizer companies and discharge from municipal and industrial
wastes. The results obtained revealed nitrate content with minimum and maximum
concentrations of 0.33 and 2.37 mg/L for RW1 and IW1 respectively as shown in Figure 6.
All measured values were below the WHO permissible value (5 mg/L). However, a fertilizer
company located at about 4km away from sample location IW1 could have contributed to the
nitrate concentration in the sample. Highly concentrated wastes containing nitrogen
compounds could have oxidized to nitrate when discharged into the environment and found
its way into ground water via percolation. Major health implications of excess nitrate in water


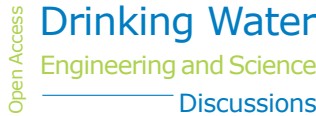

234 are hypertension in adults (Mkadmi et al. 2018) and methaemoglobinaemia in infants

235 (Bruning-Fann et al. 1993).



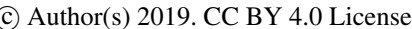


239 **Figure 6:** Spatial Variation of Nitrate in Water Samples


241 **Sulphate (SO$_4^-$)**

242 Sulphates are oxidation results of (1) their ores and (2) $H_2S$ by some bacteria activities such

243 as chlorothiobacteria and rhodothiobacteria. Their ions exist naturally in water with little or

244 no health implications recorded so far. The respective sulphate concentrations obtained for

245 IW1, IW2, RW1 and RW2 were 63, 50, 16 and 13 mg/L as shown in Figure 7. The minimum

246 concentration of 13 mg/L was revealed by water sample taken at location RW2 while

247 maximum concentration of 63 mg/L was obtained for water sample taken at location IW1.

248 All values were below the WHO, NSDWQ, EPA and IS 10500-2012 permissible values of

249 400, 100, 250 and 200 mg/L respectively. However, accumulation of sulphate in water may

250 lead to increase in water pH causing acidosis (Asamoah et al. 2011). No other health

251 implication and side effects have been recorded so far for excess sulphate in water.

252





253

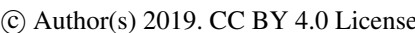

254       **Figure 7:** Spatial Variation of Sulphate in Water Samples

255

256  **Phosphate (PO$_4^-$)**

257  Phosphate is oxidized form of phosphorus which is an important nutrient for plant growth in

258  aquatic environment. Naturally, phosphate retain ability of soil coupled with its native

259  minerals low solubility enable phosphorus to be present in ground water even at very low

260  concentration (APHA, 2012). The results obtained revealed phosphate concentration varying

261  between minimum and maximum values of 0.09 and 0.19 mg/L for water sample locations

262  IW2 and IW1 respectively (Figure 8). All measured values were below WHO permissible

263  value of 6.5 mg/L. No medical implication has been reported for high concentration level of

264  phosphate in water.


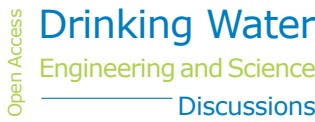

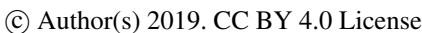


**Figure 8:** Spatial Variation of Phosphate in Water Samples

**Heavy Metals**
***Copper:*** From the result obtained (Figure 9a), minimum and maximum copper concentration
of 0.18 mg/L and 0.44 mg/L were exhibited by RW1 and IW1 respectively. However, all
values obtained were below the permissible concentration of 1 mg/L specified by NSDWQ
and WHO. When excess copper is present in water (above the permissible level),
gastrointestinal disorder occurs after long period of exposure.
***Zinc:*** Research has recorded approximately 0.05 g/Kg of zinc to be present naturally in the
earth crust (Dohare et al., 2014). From the study areas, the maximum and minimum zinc
levels were 0.911 mg/L (RW2) and 0.182 mg/L (RW1) (Figure 9b). All examined samples
revealed zinc concentration below the permissible standard values of 3, 5 and 15 mg/L set by
NSDWQ, IS 10500-2012 and WHO respectively. This could be that the zinc in its natural
mineral form (sphalerite) did not dissolve into underground water bodies via leaching in all
examined locations (Broadly et al., 2007). However, medical experts have reported
electrolyte imbalance, vomiting, acute renal failures and abdominal pain as symptoms of
excessive exposure of human to zinc.
***Lead:*** Of all heavy metals, lead is the most significant due to its toxic and harmful instinct
even at very small concentration (Gregoriadou et al., 2001). It can accumulate in body tissue
posing threat to human health. From the examined samples at different locations,
concentration of lead ranges from minimum and maximum concentrations of 0.082 mg/L
(IW2 and RW1) and 0.374 mg/L (IW1) respectively (Figure 9c). Lead concentrations of all



samples were above the permissible value of 0.01 mg/L indicated by WHO and NSDWQ.
Due to the toxic nature of lead, EPA permissible level is zero mg/L. High concentrations of
lead in samples located at IW1 and IW2 could be attributed to (1) discharge of lead-rich
waste effluents from nearby paint industry deposited in the soil which later found its way into
underground water via leaching and (2) dissolution by heavy rain of emitted aerosols and
dusts into the soil from industrial heavy plants which are transported by wind. The major
influence of high lead concentrations in water samples from RW1 and RW2 could be from
(1) leaching of natural deposits of lead ores in the soil into the groundwater (Imam, 2012) (2)
higher volume of leaded gasoline exhausts from motor vehicles in the residential area and (3)
reaction of water with removed coated-lead from pipe's surface due to turbulent motion of
transporting water from ground level to surface level. Presence of lead in water beyond
permissible level could result to hypertension, interference with Vitamin D and calcium
metabolism, brain development hindrance in foetus and young children, damage to tissues
and organs in human and many more.
***Manganese:*** Manganese is ores and rocks constituent which is widely distributed naturally. It
is a vital element for biological systems whose chemical behaviour is a function of pH,
oxidation and reduction reactions (Shand et al. 2007). The concentration of manganese in
examined samples ranged from 0.079 mg/L (IW2) to 0.481 mg/L (RW2) (Figure 9d). All
water samples exhibited manganese concentration above the permissible value of 0.05 mg/L
specified by WHO. This observation could be due to (1) ground water contact with dissolved
soil, rock and minerals of manganese in the aquifer for all sample locations (2) leaching of
industrial effluents discharge into the soil for sample locations IW1 and IW2 and (3) leachate
from landfill and sewage deposited over time in residential locations for RW1 and RW2.
Effects of high manganese concentration in water include (1) metallic and unpleasant taste to
water (2) blackish staining of laundry and plumbing fixtures and (3) formation of darkish
scales in water pipes (Takeda, 2003). However, no record of excess manganese health risk
has been recorded in human.
***Iron:*** Like manganese, iron exists in its natural form as ores (magnetite, taconite and
hematite) in rocks, soil and minerals making about 5% of the Earth's crust (Colter et al.
2006). It is dark-gray in colouration when in pure form and exists in ground water as ferric
hydroxide. Minimum and maximum iron concentrations of 0.15 and 3.26 mg/L were
observed in RW2 and IW1 respectively (Figure 9e). From the analyzed samples, two of the
samples (IW1 and RW1) have Fe concentrations above the permissible WHO, EPA, NSDWQ



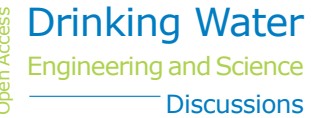

and IS 10500-2012 standard value of 0.3 mg/L with maximum concentration level revealed
by water sample collected from location IW1. The observed Fe concentration above
permissible level could be linked to (1) weathering of minerals and rocks (mineralogical and
piezometry features) of iron in the soil for the examined locations (IW1 and RW1) and (2)
dissolution of iron natural deposits into ground water bodies via leaching. However, anemia
has been reported as result of iron shortage in human. Results of extensive consumption of
drinking water containing high Fe concentration level are haermosiderosis (liver-damage
disease), diabetes mellitus, arteriosclerosis and many other neurodegenerative diseases
(Nagendrappa et al. 2010; Brewer 2009).
*Cobalt:* Cobalt can hardly be found in its native state but exists in sulphides and arsenides
form as minerals which are linnaet ($Co_3S_4$), cobaltite (CoAsS), karrolit ($CuCo_2S_4$) and
smaltyn ($CoAs_2$) (Turekian et al. 1994). In the examined sample locations, respective
minimum and maximum Co concentrations were 0.018 mg/L (RW1) and 0.073 mg/L (IW1)
(Figure 9f). Presence of cobalt could be attributed to heavy metals presence in industrial
waste effluent discharges (for IW1 and IW2) while presence in all samples could result from
leached minerals of cobalt present in the soil into underground water. No permissible
concentration of cobalt has been specified by WHO and some global agencies. It plays a key
role in the synthesis of vitamin B-12 which is an essential vitamin in human's body.
However, people exposed to high concentration of cobalt have been reported to have lungs
diseases such as wheezing, asthma and pneumonia (Chaney, 1982).
*Cadmium:* Cadmium exists as (1) natural ores in rocks and soils; and (2) zinc refining by-
product (Wang et al. 2006). Presence of cadmium in ground water occurred via leaching
when in contact with soil contaminated with discharges from mining, paints, electroplating,
petrochemical, plastics and fertilizer industries (DeZuane, 1997). Out of the examined
samples from different locations, only three (IW1, IW2 and RW1) exhibited presence of Cd
with minimum concentration of 0.001 mg/L (RW1) and maximum concentration of 0.0025
mg/L (IW2) as shown in Figure 9(g). Though Cd concentrations were below the permissible
value (0.003 mg/L) specified by WHO and NSDWQ, epidemiological studies have shown
that long-term exposure to Cd could cause (1) kidney damage (2) lung cancer (3) high blood
pressure and (4) bone defects (osteoporosis and osteomalacia). Presence of cadmium in
examined samples could be attributed to (1) leaching of waste runoff from battery industry
located at about 2.5 km away from sample location (IW1) into the soil (2) leaching of waste
discharge from paint industry located few kilometres away sample location (IW2) and (3)





galvanized steel pipe corrosion used in conveying water from the ground level to surface
level (RW1) (El-Harouny et al. 2009).
*Chromium:* Chromium exists naturally as element in rocks, soil, plants, animals and
volcanoes emissions. It is found in drinking water in trivalent (chromium 3) and hexavalent
(chromium 6) principal forms. Only IW2 and RW2 exhibited minimum and maximum Cr
concentrations of 0.0014 mg/L and 0.0022 mg/L respectively (Figure 9h). Natural deposits
erosion and coatings removal from water pipes could have been the major causatives of Cr
presence in ground water samples. Though Cr concentrations were below the WHO and
NSDWQ permissible value of 0.05 mg/L, health implications of excessive exposure to
chromium are as stated for cadmium.

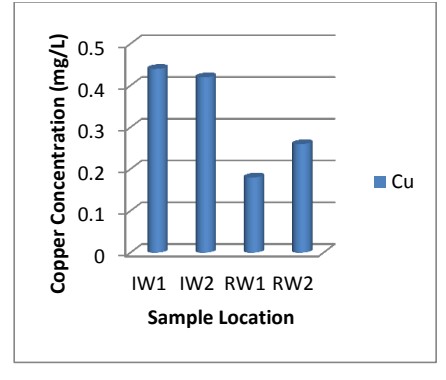

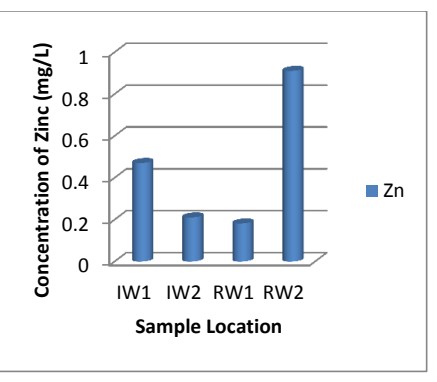


**(a)**                      **(b)**

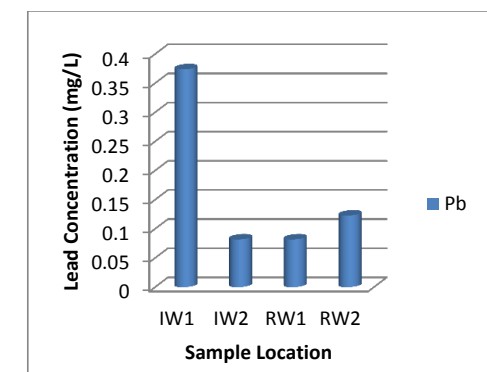

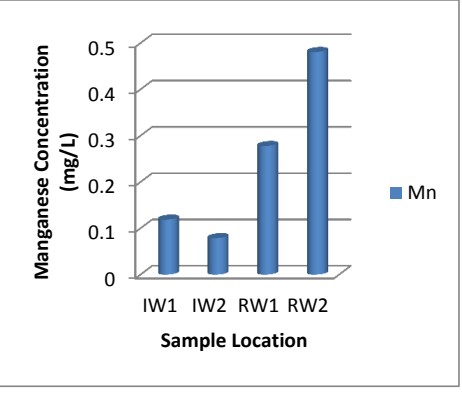


**(c)**                      **(d)**




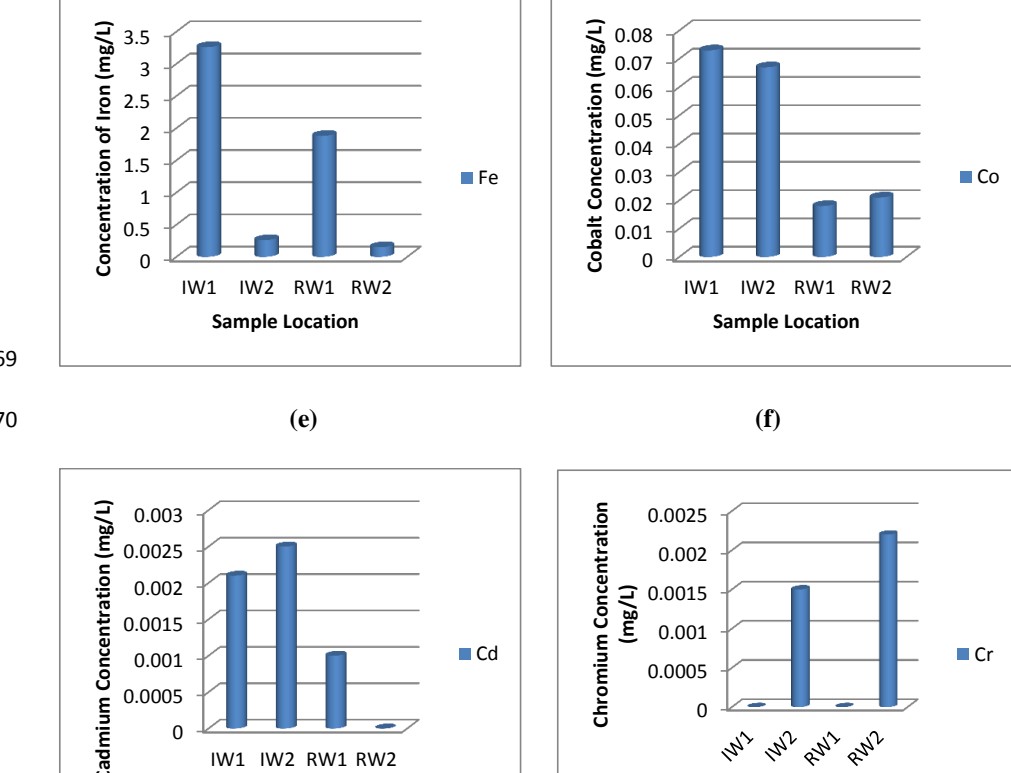


**(e)**                                    **(f)**


**(g)**                                    **(h)**

**Figure 9:** Spatial Variation of (a) Copper (b) Zinc (c) Lead (d) Manganese (e) Iron (f) Cobalt
(g) Cadmium (h) Chromium in Water Samples

**Statistical Correlation of Ground Water Contaminants**
Pearson's correlation (r) reveals existing interaction between minimum of two continuous
variables with values ranging between -1 to +1. This statistical tool was used to correlate
ground water contaminants in examined locations. A negative value implies negative
correlation while a positive value implies positive correlation between variables. A value of r
= 0 is an indication of negligible connection between parameters. In most cases, strong
correlation exists within parameters when r > 0.7 while moderate correlation exists when r
ranges between 0.5 – 0.7 (Saleem et al. 2012). Table 1 presents the Pearson's correlation



results of physico-chemical parameters of assessed water samples. The result revealed
approximately 32%, 10% and 58% of the physico-chemical parameters to be strongly ($r \geq$
0.7), moderately ($0.5 < r < 0.7$) and poorly ($r < 0.5$) correlated. At $p < 0.05$, parameters that
correlated positively with one another include: TDS with pH ($r = 0.894$) and Cu ($r = 0.805$);
pH with Cu ($r = 0.818$), Co ($r = 0.843$) and Cd ($r = 0.812$); EC with Fe ($r = 0.878$) and Cr ($r =$
0.842); COD with Co ($r = 0.808$); BOD with Co ($r = 0.8133$); $SO_4^-$ with Cu ($r = 0.886$); Fe
with Cr (0.805) and lastly Co with Cd ($r = 0.821$). At $p < 0.01$, parameters that strongly
correlated with one another include: TDS with Co ($r = 0.947$) and Cd ($r = 0.956$); COD with
BOD ($r = 0.999$) and Cu ($r = 0.949$); BOD with Cu ($r = 0.956$); $NO_3^-$ with $PO_4^-$ ($r = 0.908$)
and Pb ($r = 0.990$); $SO_4^-$ with Co ($r = 0.980$); $PO_4^-$ with Pb ($r = 0.926$); Cu with Co ($r =$
0.932); and lastly Mn with Cd ($r = 0.987$). Though none of the remaining parameters was
negatively correlated, they were poorly significantly correlated with r values of less than 0.7
at $p < 0.05$ or $< 0.01$. However, majority of the measured physico-chemical parameters
correlated with one another at either $p < 0.05$ or $< 0.01$ which is an indication that availability
of specified pollution indicators will definitely have influence on other assessed pollutants in
water samples located at both industrial (IW1 and IW2) and residential (RW1 and RW2)
locations.
**Table 1:** Pearson's correlation of Physico-Chemical Parameters of Water Samples

|     | TDS | pH | EC | COD | BOD | Cl⁻ | NO₃⁻ | SO₄⁻ | PO₄⁻ | Cu | Zn | Pb | Mn | Fe | Co | Cd | Cr |
|-----|-----|-----|-----|-----|-----|-----|-----|-----|-----|-----|-----|-----|-----|-----|-----|-----|-----|
| **TDS** | 1.00 | | | | | | | | | | | | | | | | |
| **pH** | **0.894** | 1.00 | | | | | | | | | | | | | | | |
| **EC** | 0.138 | 0.003 | 1.00 | | | | | | | | | | | | | | |
| **COD** | 0.612 | 0.627 | 0.059 | 1.00 | | | | | | | | | | | | | |
| **BOD** | 0.623 | 0.649 | 0.049 | **0.999** | 1.00 | | | | | | | | | | | | |
| **Cl⁻** | 0.063 | 0.050 | 0.084 | 0.597 | 0.574 | 1.00 | | | | | | | | | | | |
| **NO₃⁻** | 0.192 | 0.071 | 0.494 | 0.545 | 0.516 | 0.775 | 1.00 | | | | | | | | | | |
| **SO₄⁻** | 0.905 | 0.729 | 0.276 | 0.787 | 0.785 | 0.261 | 0.481 | 1.00 | | | | | | | | | |
| **PO₄⁻** | 0.026 | 0.001 | 0.522 | 0.262 | 0.237 | 0.698 | **0.908** | 0.205 | 1.00 | | | | | | | | |
| **Cu** | **0.805** | **0.818** | 0.059 | **0.949** | **0.956** | 0.374 | 0.398 | **0.886** | 0.135 | 1.00 | | | | | | | |
| **Zn** | 0.270 | 0.153 | 0.120 | 0.015 | 0.013 | 0.392 | 0.063 | 0.102 | 0.137 | 0.008 | 1.00 | | | | | | |
| **Pb** | 0.180 | 0.051 | 0.593 | 0.466 | 0.437 | 0.693 | **0.990** | 0.460 | **0.926** | 0.340 | 0.036 | 1.00 | | | | | |
| **Mn** | 0.913 | 0.716 | 0.226 | 0.332 | 0.339 | 0.003 | 0.098 | 0.754 | 0.005 | 0.534 | 0.547 | 0.107 | 1.00 | | | | |
| **Fe** | 0.001 | 0.088 | **0.878** | 0.005 | 0.009 | 0.024 | 0.299 | 0.039 | 0.449 | 0.010 | 0.056 | 0.395 | 0.031 | 1.00 | | | |
| **Co** | **0.947** | **0.843** | 0.163 | **0.808** | **0.813** | 0.219 | 0.369 | **0.980** | 0.118 | **0.932** | 0.106 | 0.340 | 0.768 | 0.004 | 1.00 | | |
| **Cd** | **0.956** | **0.812** | 0.157 | 0.402 | 0.412 | 0.003 | 0.094 | 0.784 | 0.002 | 0.615 | 0.474 | 0.096 | **0.987** | 0.006 | **0.821** | 1.00 | |
| **Cr** | 0.097 | 0.002 | **0.842** | 0.002 | 0.004 | 0.013 | 0.135 | 0.129 | 0.157 | 0.005 | 0.405 | 0.208 | 0.265 | **0.805** | 0.065 | 0.174 | 1.00 |







**Conclusion**

This present study examined ground water samples from two different industrial and residential locations of Lagos metropolis for some selected physico-chemical parameters which include: TDS, pH, EC, COD, BOD, $Cl^-$, $NO_3^-$, $SO_4^-$, $PO_4^-$, Cu, Zn, Pb, Mn, Fe, Co, Cd and Cr. From the executed laboratory analysis for ground water samples from industrial locations, all measured values of physico-chemical parameters were either below permissible values or within ranges specified by Nigerian Standard for Drinking Water Quality and World Health Organization except pH, TDS, EC, Pb, Mn and Fe while only pH, Pb, Mn and Fe violated permissible values for ground water samples collected from residential locations. From the result obtained, higher concentrations of TDS and EC were reported for ground water samples collected from industrial locations than those from residential locations due to heavy discharge of effluents from industrial treatment plants as well as dissolution of ionic heavy metals from industrial activities of heavy machines. Thus, ground water samples from residential locations are more suitable for drinking than those from industrial locations. Also, the statistical Pearson's correlation result revealed measured physico-chemical parameters to be moderately and strongly correlated with one another at either $p < 0.05$ or $< 0.01$.

**Recommendations**

However, due to presence of higher concentrations of Pb, Mn and Fe in all ground water samples, a low cost water treatment with chlorine should be employed to enhance transformation of the metals into solid settlement which can be filtered out before drinking. Also, blood samples of residents drinking samples of ground water collected from locations should be examined for future research work to know the levels of Pb, Mn and Fe in their blood streams. This will enable them to know their health status in this regard and also help medical experts in the field to recommend drugs if need be for residents exposed to excess concentrations of these heavy metals.



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
