# Peer review of "Industrial and Residential Ground Water Physico-Chemical Properties Assessment in Lagos Metropolis"

_Drinking Water Engineering and Science, 2019_

## Referee Comment (RC1) · Anonymous Referee #1 · 26 Mar 2019

General comments The manuscript gives information about the physico-chemical properties in Lagos Metropolis. However, the methodology used, the sampling period, the presentation of the results and the no consideration of the aquifer characteristics, local/regional groundwater flow, etc., do not allow to have a proper understanding of the problem.

Methodological synthesis - A location figure is missing, indicating the specific location of the study area (i.e., boreholes, Deli foods, OK foods, Lagos lagoon, fertilizer company (for Nitrate concentrations), etc.) - The description of the aquifer characteristics is a must. Does it correspond to a phreatic, semi-confined or confined aquifer? Please

provide the lithology of the study area, including the depth of the boreholes used during the research. Do the boreholes correspond to the same aquifer layer? - Has any vulnerability assessment been conducted in the study area? It may provide more insights for the analysis of the present results. - How many samples were extracted? - The sampling protocol was too limited. Seasonal behavior should have addressed in order to establish a groundwater assessment. This issue really constraints the scope of this study.

Results and Discussion - A wrong type of plots has been selected by the authors. Box plots, or a similar method, must be used for the results assessment. The number of samples is missing. - The spatial distribution of the sampling points, including the location of facilities of interest, avoids conducting a proper analysis of the results. - Were the samples extracted during rainy season? No information has been provided in order to support the statement related to Nitrate (the authors say: The alkalinity of water sample (IW2) may be attributed to the presence of bicarbonates, part of essential raw materials for production, lost into the soil and percolates into the underground soil via rain water).

Conclusions Local groundwater flow information should have been considered in this study to make it valid. No information regarding to the lithology of the aquifer and potential geogenic contamination of the groundwater has been provided, which limits the assessment and conclusions of this study. Although the Pearson's correlation for some parameters seems to be conclusive, the data analysis and the methodology is limited.

Please also note the supplement to this comment:
https://www.drink-water-eng-sci-discuss.net/dwes-2019-2/dwes-2019-2-RC1-supplement.pdf

---

## Author Comment (AC1) · 22 Apr 2019

**Response to Referees' Queries (RC1-DWES)**

| Referee's Queries | Authors' Responses |
|---|---|
| **RC1** | |
| 1. - A location figure is missing, indicating the specific location of the study area (i.e., boreholes, Deli foods, OK foods, Lagos lagoon, fertilizer company (for Nitrate concentrations), etc. | These have been provided in Figure 1. |
| 2. - The description of the aquifer characteristics is a must. Does it correspond to a phreatic, semi-confined or confined aquifer? Please provide the lithology of the study area, including the depth of the boreholes used during the research. Do the boreholes correspond to the same aquifer layer? | Aquifer characteristics of study areas has been provided as obtained from Lagos State water regulatory commission together with the lithology information stated thus "The aquifer in all the study areas was characterized as unconfined under saturated zone which is coastal plain sand comprising of silt, clay and sand (Akinlalu et al. 2018) while the respective borehole depths for Deli foods, OK foods, Oshodi and Agege community boreholes were 117, 134, 109 and 143 m (www.lswrc.lagosstate.gov.ng)." |
| 3. - Has any vulnerability assessment been conducted in the study area? It may provide more insights for the analysis of the present results. | Results of previous research works involving groundwater quality and vulnerability assessment have been presented in Table 1 to shed more lights on the present study. |
| 4. - How many samples were extracted? | Total number of samples extracted was four (4). This has been specified already in Samples collection sub-section thus "A total of four samples were collected from the locations." |
| 5. -The sampling protocol was too limited. Seasonal behavior should have addressed in order to establish a groundwater assessment. This issue really constraints the scope of this study. | We totally in agreement with the reviewer in this regard. Seasonal behaviour in terms of both rainy (wet) and sunny (dry) seasons should have been considered to expand the scope of the study. Thereby, expanding results discussion in terms of comparison. However, samples collection was executed during the rainy season as specified thus "This sample collection exercise was done during rainy season (May 2018) and ………….". Our future research works will put that into consideration. |
| 6. - A wrong type of plots has been selected by the authors. Box plots, or a similar method, must be used for the results assessment. The number of samples is missing. | The wrong type of plots have been deleted and replaced with Table 1 comprising of all the information earlier presented in figures and number of samples examined. |
| 7. - The spatial distribution of the sampling points, including the location of facilities of interest, avoids conducting a proper analysis of the results. | All these information has been provided under the "samples collection" and "materials and methodology sections" thus "All chemicals and reagents (sodium hydroxide, distilled water, buffer solution, indicators, silver nitrate, sodium trioxocarbonate IV, ethylenediamminetetraacetic acid (EDTA), bleaching powder, potassium iodide, sodium thiosulphate solution, manganese sulphate and hydrochloric acid) used for |

| | laboratory analysis were of analytical grade and purchased from TopJ Scientific in Ado-Ekiti, Ekiti State, Nigeria. Laboratory analysis was conducted inside the waste water treatment laboratory of Afe Babalola University, Ado-Ekiti, Ekiti state." and "Two litres of groundwater samples were collected from boreholes of Deli foods (IW1) and OK foods (IW2). Also, groundwater samples were collected from Oshodi and Agege community boreholes, each located at 40 km away from Deli foods (RW1) and OK foods (RW2) respectively. A total of four samples were collected from the locations." |
|---|---|
| 8. -Were the samples extracted during rainy season? No information has been provided in order to support the statement related to Nitrate (the authors say: The alkalinity of water sample (IW2) may be attributed to the presence of bicarbonates, part of essential raw materials for production, lost into the soil and percolates into the underground soil via rain water). | Thanks very much for this notification. This information was formerly presented technically in the manuscript where the period of collection was indicated to be rainy season period (May 2018). However, it has been signified that the period of sample collection was rainy season. This was used to validate the discussion of result identified. |
| 9. Local groundwater flow information should have been considered in this study to make it valid. No information regarding to the lithology of the aquifer and potential geogenic contamination of the groundwater has been provided, which limits the assessment and conclusions of this study. Although the Pearson0s correlation for some parameters seems to be conclusive, the data analysis and the methodology is limited. | Information regarding the aquifer lithology has been provided under the "study area" section to make the local groundwater flow information available thus: "The aquifer in all the study areas was characterized as unconfined under saturated zone which is coastal plain sand comprising of silt, clay and sand (Akinlalu et al. 2018) while the respective borehole depths for Deli foods, OK foods, Oshodi and Agege community boreholes were 117, 134, 109 and 143 m". |

---

## Author Comment (AC2) · 22 Apr 2019

**Response to Referees' Queries (DWES-RC2)**

| RC2 | |
|---|---|
| 1. It is not clear how this research relates to existing literature. | 1. This study has been related with previous studies to check its correlation with existing literatures by presenting values of previous studies with the present study in Table 1. |
| 2. It is not clear from the paper why only these physico-chemical parameters were chosen (and not microbiological). | Reason for using these properties has been supported with a reference thus: "In this study, groundwater samples were obtained from prominent industrial and residential locations of Lagos State, Nigeria and were evaluated to know the concentration of heavy metals (copper, zinc, lead, manganese, iron, colbalt, cadmium and chromium) that are present and other physico-chemical properties such as total dissolved solid, pH, electrical conductivity, chemical oxygen demand, biological oxygen demand, chloride, nitrate, sulphate and phosphate (Mohsin et al. 2013)." |
| 3. Only insert figures when it adds to the information in the text. This is not the case for the Figures presented in the manuscript. All information could be given in one Table. | All information has been presented in one Table (Table 1) as instructed by the referee. |
| 4. Ground water = groundwater. | This has been taken care of. |
| 5. 22-23, avoid conclusions that cannot be drawn from the data, since microbiological parameters are also of importance for drinking water production. In addition, the risk of future contaminations is present in all urban groundwaters in the world. | This conclusion has been removed. |
| 6. - 30-31, not clear what is meant. | The statement "Due to these attributes, enormous waste effluents are being generated on an hourly basis through industrial and residential activities with higher demand for domestic water consumption linked to her densely populated instinct." means high industrial and manufacturing activities of Lagos contributed to contamination of groundwater due to large waste effluents generated from these industries. |
| 7. - 66, "to know the concentration of heavy metals" | This has been corrected. |
| 8. - 82, not clear what is meant and give reference | This gives brief information about the study area. Reference has been cited. |
| 9. - 89-90, not clear what is meant. | The sentence has been reframed for proper understanding thus: "The high population with enormous commercial activities attributes was the major reason for choosing these industrial and residential areas as case studies for this research work." |
| 10. - 92, delete "each" | The word "each" has been deleted. |
| 11. - 104-105, insert reference | Reference has been inserted. |
| 12. - 115, DO is not unitless | Thank you for this huge notification. This has been corrected accordingly. |
| 13. - 124-125, explanation of pH is not necessary to give. | The explanation of pH has been removed. |

| | |
|---|---|
| 14. - 127-128, delete sentence (repetition) | This sentence "This revealed pH range of 6.35 to 7.58 with minimum and maximum exhibited by RW1 and IW1 respectively" has been deleted as instructed. |
| 15. - 135, this explanation is highly disputable. $CO_2$ in groundwater does not come from the atmosphere. | This statement has been restructured for better understanding thus "Slightly acidic nature of RW1 and RW2 may result from the deposit carbonic acid formed via reaction of carbon dioxide (from larger population in residential areas than industrial areas) with rain water (Tiwari et al, 2015)." |
| 16. - 138, which chemical processes? | An example of chemical process has been given thus: "This may be transported from soil surface level to form deposits in the ground water via chemical process such as leaching over period of time." |
| 17. - 150-152, not necessary to explain TDS. Combine TDS with EC. | TDS explanation has been removed. |
| 18. - 171-172, delete sentence (repetition) | This sentence "The values range between 450-1190 μS/cm with IW1 exhibiting maximum EC while RW2 exhibited minimum EC." has been removed. |
| 19. - 192, avoid conclusions that cannot be drawn from the data, since microbiological parameters are also of importance for drinking water production. In addition, the risk of future contaminations is present in all urban groundwaters in the world. | This conclusion has been restructured to reflect better understanding. |
| 20. - 194-197, how this conclusion can be drawn? | Yes, this conclusion can be drawn as increased volume of organics from municipal and solid wastes from residential areas are major contributor of both chemical and biological oxygen demand. |
| 21. - 225-227, not necessary to explain nitrate, etc | The statement has been removed. |
| 22. -397-399, what is the meaning of these correlations? Can conclusions be drawn about the source of the pollution? | The correlation does not give information about the source of pollution. It only gives information about the level of dependency of pollutants over one another. |
| 23. - 416-419, avoid conclusions that cannot be drawn from the data, since microbiological parameters are also of importance for drinking water production. In addition, the risk of future contaminations is present in all urban groundwaters in the world. | The conclusions were arrived at based on the examined parameters excluding micobiological parameters. Thus the statement has been reframed thus: "With reference to the examined properties, groundwater samples from residential locations are more suitable for drinking than those from industrial locations." |
| 24. - 423, low cost treatment with chlorine will not remove Fe, Mn and Pb. | Authors are not saying chlorine will remove Fe, Mn and Pb. The statement reflects transformation of the heavy metals into sediments which will enhance their removal via filtration due to the presence of added chlorine. |
| 25. - 425-429, not necessary to do this research when it is evident that water quality is bad, so more emphasis must be given to centralized water supply and sewerage. | This recommendation was made to enable researchers expose the world to level of exposure of residents to heavy metals contamination on time basis. A mathematical model can be developed in future to forecast the rate of exposure. |

---

## Author Comment (AC4) · 22 Apr 2019

The comment was uploaded in the form of a supplement:
https://www.drink-water-eng-sci-discuss.net/dwes-2019-2/dwes-2019-2-AC4-supplement.pdf